# Long-Term Impact of D2 Lymphadenectomy during Gastrectomy for Cancer: Individual Patient Data Meta-Analysis and Restricted Mean Survival Time Estimation

**DOI:** 10.3390/cancers16020424

**Published:** 2024-01-19

**Authors:** Alberto Aiolfi, Davide Bona, Gianluca Bonitta, Francesca Lombardo, Michele Manara, Andrea Sozzi, Diana Schlanger, Calin Popa, Marta Cavalli, Giampiero Campanelli, Antonio Biondi, Luigi Bonavina

**Affiliations:** 1I.R.C.C.S. Ospedale Galeazzi–Sant’Ambrogio, Division of General Surgery, Department of Biomedical Science for Health, University of Milan, Via C. Belgioioso, 173, 20157 Milan, Italy; davide.bona@unimi.it (D.B.); bbonit@icloud.com (G.B.); francesca.lombardo89@gmail.com (F.L.); michele.mnra@gmail.com (M.M.); sozzi94@hotmail.it (A.S.); 2Surgery Clinic 3, Regional Institute of Gastroenterology and Hepatology “Prof. Dr. Octavian Fodor”, “Iuliu Hațieganul” University of Medicine and Pharmacy, 400394 Cluj-Napoca, Romania; schlanger.diana@yahoo.com (D.S.); calinp2003@yahoo.com (C.P.); 3I.R.C.C.S. Ospedale Galeazzi-Sant’Ambrogio, Division of General Surgery, Department of Surgery, University of Insubria, 20157 Milan, Italy; marta_cavalli@hotmail.it (M.C.); giampiero.campanelli@grupposandonato.it (G.C.); 4Department of General Surgery and Medical Surgical Specialties, G. Rodolico Hospital, Surgical Division, University of Catania, 95131 Catania, Italy; abiondi@unict.it; 5Department of Biomedical Sciences for Health, Division of General and Foregut Surgery, IRCCS Policlinico San Donato, University of Milan, 20097 Milan, Italy; luigi.bonavina@unimi.it

**Keywords:** gastric cancer, lymphadenectomy extent, D2 lymphadenectomy, overall survival, cancer specific survival, disease-free survival

## Abstract

**Simple Summary:**

Debate exists regarding the effect of D2 vs. D1 lymphadenectomy on long-term oncological outcomes after gastrectomy for cancer. The aim of our individual patient data meta-analysis was to assess the impact of D2 lymphadenectomy on long-term survival after gastrectomy for gastric cancer. Compared with previous randomized trials, we confirmed a trend toward improved overall survival (1.8 months; 95% CI −4.2, 0.7; *p* = 0.14), cancer specific survival (1.2 months, 95% CI −3.9, 5.7; *p* = 0.72), and disease-free survival (0.8 months, 95% CI −1.7, 3.4; *p* = 0.53) at 60-month follow-up for D2 lymphadenectomy. Compared to D1, D2 lymphadenectomy is associated with a clinical trend toward improved OS, CSS, and DFS at 60-month follow-up.

**Abstract:**

Background: Debate exists concerning the impact of D2 vs. D1 lymphadenectomy on long-term oncological outcomes after gastrectomy for cancer. Methods: PubMed, MEDLINE, Scopus, and Web of Science were searched and randomized controlled trials (RCTs) analyzing the effect of D2 vs. D1 on survival were included. Overall survival (OS), cancer-specific survival (CSS), and disease-free survival (DFS) were assessed. Restricted mean survival time difference (RMSTD) and 95% confidence intervals (CI) were used as effect size measures. Results: Five RCTs (1653 patients) were included. Overall, 805 (48.7%) underwent D2 lymphadenectomy. The RMSTD OS analysis shows that at 60-month follow-up, D2 patients lived 1.8 months (95% CI −4.2, 0.7; *p* = 0.14) longer on average compared to D1 patients. Similarly, 60-month CSS (1.2 months, 95% CI −3.9, 5.7; *p* = 0.72) and DFS (0.8 months, 95% CI −1.7, 3.4; *p* = 0.53) tended to be improved for D2 vs. D1 lymphadenectomy. Conclusions: Compared to D1, D2 lymphadenectomy is associated with a clinical trend toward improved OS, CSS, and DFS at 60-month follow-up.

## 1. Introduction

Surgical resection is the cornerstone of gastric cancer (GC) treatment [1,2,3,4,5]. Lymph node (LN) involvement is one of the most important survival prognostic factors, but the extent of lymphadenectomy remains debated. The degree of LN involvement depends on tumor location, submucosal infiltration, differentiation, tumor size, and ulceration [6]. D2 lymphadenectomy (D2) is generally accepted as the standard procedure during curative gastrectomy [2,3]. In contrast with Eastern countries, where D2 has been performed with satisfying outcomes for decades, D1 lymphadenectomy (D1) is more commonly performed in the West [7]. The theoretical benefit of D2 is based on an increased number of resected nodes which implies more accurate staging, the removal of potential metastatic lymph nodes, and a reduction in the risk of loco-regional recurrence [8,9]. The guidelines of the 8th International Union for Cancer Control/American Joint Committee on Cancer (IUCC/AJCC) suggest that >15 LNs are needed for reliable staging [10]. The effect of D2 on survival has been proven to be beneficial in numerous observational studies [11,12,13,14,15,16,17,18]; however, available evidence from randomized controlled trials does not univocally support the routine implementation of D2 in GC patients. The optimal extent of LN dissection in patients with GC necessary to avoid understaging/undertreatment, minimize related complications, and improve survival is a matter of debate.

The aim of this review was to assess the influence of D2 on long-term survival after curative gastrectomy for cancer using a multivariate method for meta-analysis of restricted mean survival time difference (RMSTD) with individual patient data (IPD).

## 2. Materials and Methods

A systematic review was conveyed following the Preferred Reporting Items for Systematic Reviews and Meta-Analyses checklist guideline (PRISMA 2020) [19]. Scopus, MEDLINE, Web of Science, ClinicalTrials.gov, Cochrane Central Library, and Google Scholar were screened [20]. The first search was run in January 2023, repeated in June 2023, and updated in October 2023. A combination of the following MeSH terms (Medical Subject Headings) was used: “cancer”, “neoplasm”, “adenocarcinoma”, “carcinoma”, “gastrectomy”, “subtotal gastrectomy”, “lymphadenectomy”, “overall survival”, “disease-free survival”, “cancer-specific survival”, and “mortality” (File S1). All titles were screened, fitting abstracts were obtained, and the reference lists were appraised by three independent authors (AA, DS, MM). The study was registered with PROSPERO (CRD42023457461). Ethical approval was not required.

### 2.1. Eligibility Criteria

The inclusion criteria were as follows: (1) randomized controlled trials (RCTs) describing long-term survival data or Kaplan–Meier curves for D2 and D1 lymphadenectomy in the setting of operable gastric cancer; (2) when two or more articles were published by the same institution, study group, or used the same data set, articles with the longest follow-up or the largest sample size were included; and (3) in cases of duplicates, the most recent study was considered. The exclusion criteria were as follows: (1) language other than English; (2) studies with non-comparative analysis for D2 vs. D1; (3) studies reporting mixed/aggregate data counting other surgical procedures (esophageal resections); (4) studies not reporting the a priori defined primary outcome (OS); and (5) studies with less than 10 patients per study arm.

### 2.2. Data Extraction

Information regarding the authors, year of publication, study designs, country, patient number, age, gender, American Society of Anesthesiologists (ASA) physical status, body mass index (BMI), comorbidities, tumor characteristics, tumor location, surgical approach, postoperative and pathological outcomes, follow-up duration and survival was collected. All data were autonomously collected by three authors (AA, DS, MM) and compared at the end of the research. A fourth author (LB) revised the database and clarified inconsistencies.

### 2.3. Outcome of Interest and Definition

The primary outcome was overall survival (OS). OS was defined as the time from surgery to last known follow-up and death. Secondary outcomes were disease-free survival (DFS) and cancer specific survival (CSS). DFS was defined as the time from surgical procedure to cancer recurrence (distant or local) or death. CSS was defined as the time from surgical procedure to death due to gastric cancer. Data on OS, DFS, and CSS were extracted using Kaplan–Meier survival curves. Gastric cancer was defined as any primary histopathological confirmed neoplasm located in the upper, middle, or lower portion of the stomach. Lymph nodes were classified into stations ranging from 1 to 16. D1 lymphadenectomy entails the dissection of perigastric lymph nodes along the lesser and greater gastric curvature (station numbers 1–6). D2 lymphadenectomy involves additionally dissecting lymph nodes around the celiac axis (station numbers 7–12). 

### 2.4. Quality Assessment and Assessment of Certainty of Evidence

Two authors (AA, DS) independently assessed the methodological quality of the selected trials by using the Cochrane risk of bias tool [21]. This tool appraises four criteria: (1) the method of randomization; (2) the allocation concealment; (3) the baseline comparability of study groups; and (4) the blinding and completeness of follow-up. Trials were classified as follows: A = adequate, B = unclear, and C = inadequate on each criterion. Thus, each RCT was graded as having a low, moderate, or high risk of bias. Disagreements were solved through a discussion. The Grading of Recommendations, Assessment, Development, and Evaluation (GRADE) tool was utilized to assess the quality of the body of evidence across studies [22]. We constructed GRADE evidence profiles of certainty for each comparison and outcome using GRADEpro GDT (https://www.gradepro.org, accessed on 10 December 2023). The certainty of evidence is determined by the risk of bias across studies, incoherence, indirectness, imprecision, publication bias, and other parameters [23]. 

### 2.5. Statistical Analysis

The results of the systematic review were summarized into a Frequentist meta-analysis of restricted mean survival time difference (RMSTD) [24,25,26]. Individual patient time-to-event data (IPD) were obtained from Kaplan–Meier curves [27]. The Get Data Graph Digitizer software version 2.26 (https://getdata-graph-digitizer.software.informer.com/, accessed on 10 December 2023) was used for curves digitalization. The calculation of pooled RMSTD was reported using a random effect multivariate meta-analysis. Additionally, using IPD, we implemented the flexible hazard-based regression model with the inclusion of a normally distributed random intercept The time-dependent effects of surgical treatment were parametrized as interaction terms between the surgical treatment and the baseline hazard and statistically tested by likelihood ratio test. The hazard functions plot was reported using marginal prediction [28]. Two-sided *p*-values were considered statistically significant when less than 0.05 and the CIs were computed at 95%. R software application (version 3.2.2; R Foundation, Vienna, Austria) was used for statistical analysis [29].

## 3. Results

### 3.1. Systematic Review

The selection process flow chart is reported in Figure 1. 

Overall, 3755 publications were identified, with 2841 titles being screened. In total, 244 abstracts and 87 full-text articles were examined after being found to possibly be relevant. After evaluation, five RCTs met the inclusion/exclusion criteria and were incorporated in the quantitative analysis. The quality of studies is depicted in Appendix A. The included RCTs had issues regarding the blinding of participants. The randomization method was detailed in all studies, while the operating surgeon proficiency was specified in one study. Details on power analysis were specified in four trials; sample size calculation was not reported in another trial in which the low number of enrolled patients raised the suspicion of no sample size calculation (Appendix A). 

Quantitative analysis was performed on an intention-to-treat basis. Overall, 1653 patients undergoing curative gastrectomy for cancer were included (Table 1). Of these, 805 (48.7%) underwent D2 lymphadenectomy. The patient age ranged from 30 to 87 years old, and the majority were males (61.2%). All patients had histologically proven gastric adenocarcinoma according to the Lauren classification. Subtotal or total gastrectomy was performed in 1045 patients (63.3%) and 608 patients (36.7%), respectively. The extension of the gastric resection was independent of randomization and was decided upon depending on tumor location, margins, and cancer histology. Pathological tumor stage according to the 5th edition of the American Joint Committee on Cancer was specified in three studies; stage I: 41.7%, stage II: 23.6%; stage III: 28%, and stage IV: 6.7%. Tumor staging according to the pTNM classification was reported in all studies; 34.2% of patients were pT3-T4, whereas 56.3% were pN+. Open resection was performed in all patients, whereas the anastomotic technique varied among studies depending on surgeon preferences. None of the patients received neoadjuvant or adjuvant chemoradiotherapy. 

### 3.2. Meta-Analysis—OS

The clinical appraisal of the RMSTD was based on studies reporting Kaplan–Meier OS curves (Appendix A). The RMSTD estimation at different time horizons is detailed in Table 2. At τ_5_ = 60 months (five studies), the combined effect from the multivariate meta-analysis is 1.8 months (95% CI −4.2, 0.7; *p* = 0.14) indicating that at 5-year follow-up, D2 patients lived 1.8 months longer on average compared with D1 patients. At τ_10_ = 120 months (three studies), the combined effect from the multivariate meta-analysis is 1.6 months (95% CI −9.2, 6.1; *p* = 0.68), indicating that at 10-year follow-up, D2 patients lived 1.6 months longer on average compared with D1 patients. The estimated pooled OS curves for D2 and D1 are depicted in Figure 2.

### 3.3. Meta-Analysis—CSS and DFS

The analysis for CSS showed that at τ_5_ = 60 months (three studies), the combined effect from the multivariate meta-analysis is 1.2 months (95% CI −3.9, 5.7; *p* = 0.72), indicating that at 5-year follow-up, D2 patients tended to have a trend toward an improved DFS compared with D1 patients. At τ_10_ = 120 months (two studies), the combined effect from the multivariate meta-analysis is 6.1 months (95% CI −5.2, 17.7; *p* = 0.28) (Table 3) (Appendix A). Similarly, the analysis for DFS showed that at τ_5_ = 60 months (three studies), the combined effect from the multivariate meta-analysis is 0.8 months (95% CI −1.7, 3.4; *p* = 0.53), indicating that at 5-year follow-up, D2 patients tended to have an improved CSS compared with D1 patients. At τ_10_ = 120 months (two studies), the combined effect from the multivariate meta-analysis is 3.5 months (95% CI −3.1, 10.2; *p* = 0.29) (Table 4) (Appendix A). The estimated pooled CSS and DFS curves for D2 and D1 are depicted in Figure 3 and Figure 4, respectively. Using the GRADE tool, the certainty of evidence for the assessed outcomes was between moderate and high because of confounding bias, inconsistency, and imprecision (Appendix A).

## 4. Discussion

This study shows that D2 lymphadenectomy seems to be associated with a more favorable clinical trend regarding OS, CSS, and DFS compared to D1. Specifically, the RMSTD analysis shows a mean survival benefit of 1.8, 1.5, and 0.8 months at 60-month follow-up, respectively. Similarly, the very-long-term OS analysis (120 months) shows a trend toward improved survival in D2 patients (1.6 months; *p* = 0.68).

The value of surgical dissection to remove the draining lymph nodes is controversial. Japanese surgeons regularly perform more extended dissections (i.e., D2 or D3) as opposed to the more limited (D1) lymph node dissection most commonly performed in Western countries. Theoretically, a more extended lymph node dissection may improve survival due to more accurate disease staging (N-stage) and increased likelihood of removing microscopic metastatic deposits [35,36,37]. Previous observational studies and meta-analyses reported promising survival benefits for D2 over D1 lymphadenectomy [9,38,39,40]. A 2015 Cochrane analysis reported no significant differences in terms of 5-year OS (HR = 0.91; 95% CI 0.71–1.17) and DFS (HR = 0.95; 95% CI 0.84–1.07), whereas a significantly improved DSS was found for D2 vs. D1 (HR = 0.81; 95% CI 0.71–0.92) [41]. Furthermore, a 2021 meta-analysis reported a trend toward improved 5-year OS in T3 (OR = 1.64; 95% CI 1.01–2.67; *p* = 0.05) and N+ patients (OR = 1.36; 95% CI 0.98–1.87; *p* = 0.06) who underwent D2 [42]. Based on these findings, the NCCN and ESMO guidelines support the use of D2 lymphadenectomy as the best curative option in patients with potentially curable gastric cancer [2,3]. The routine utilization of extended lymph nodes dissection is associated with higher postoperative morbidity and mortality, even in the case of spleen-preserving resections, and with the lack of clear survival benefits in most large RCTs [30,31,33,34,35]. Notably, the 2015 Cochrane DSS analysis was associated with moderate heterogeneity (I^2^ = 40%), whereas the sensitivity analysis including only the European trial showed no statistically significant association between lymphadenectomy extent and DSS (HR = 0.86; 95% CI 0.72–1.01; I^2^ = 0%) [39]. Therefore, robust and definitive conclusions regarding the real benefits of D2 lymphadenectomy on long-term OS, CSS, and DFS are missing [43].

In our study, D2 has a modest clinical impact on long-term (5-year) OS. This is supported by the RMSTD analysis that demonstrated an average OS improvement of 1.8 months (95% CI −4.2, 0.6). Our findings are different from those of Wu et al., who found a higher 5-year OS for D2 vs. D1 (59.5% vs. 53.6%, *p* = 0.041) with a significantly reduced risk of mortality (HR = 0.49; *p* = 0.002) [32]. In contrast, our findings are in line with the IGCSG trial, which reported non-significantly improved 5-year OS (HR = 0.82; *p* = 0.35) [34]. Also, Cuschieri et al. (HR = 1.1, 95% CI 0.87–1.39) and Songun et al. (HR = 0.92, 95% CI 0.78–1.09) conveyed no significant difference in the 5-year OS [31,33]. Interestingly, the RMSTD very-long-term OS analysis (120 months) showed a clinical trend toward improved OS in D2 patients (1.5 months, 95% CI −9.2, 6.1; *p* = 0.68). This result is similar to those of Degiuli et al. (HR = 0.98; *p* = 0.94) and Songun et al. (29% vs. 21%; *p* = 0.34), who found no significant differences but a modest clinical improvement with very-long-term OS for D2 vs. D1 [33,34]. The RMSTD CSS estimation assessed that D2 patients had a trend toward a modest but statistically nonsignificant benefit compared to D1. This is similar to Degiuli et al. [34], who described non-significant 5-year CSS differences (HR = 1.02; *p* = 0.91), but in contrast with Wu et al. (HR = 0.7; *p* = 0.006) and Songun et al. (HR = 0.74), who found significantly improved CSS in D2 patients [32,33]. Finally, in accordance with Cuschieri et al. [31], Wu et al. [32], and the Dutch trial [33], no significant differences were estimated for 5-year DFS. 

Notably, some important issues should be considered while interpreting our results. First, in the included trials, patients did not receive neoadjuvant or adjuvant treatments. Considering the current adoption of multimodal treatments in resectable GC patients (stage IB or greater), the “real-world” applicability of our results should be defined. Specifically, it remains unclear whether the hypothetical survival advantage associated with more extended lymphadenectomy can be synergically combined with that of adjuvant/neoadjuvant treatments or whether this might make extended lymph node dissection gratuitous [1,44]. In this light, it has been reported that induction treatments affect lymph node yield, with a significant reduction in the mean number of harvested lymph nodes (29.6 vs. 25.3, *p* = 0.002) and a higher proportion of patients with <15 lymph nodes harvested (24.1% vs. 7.7%) [45]. Tissue fibrosis/sclerosis with difficult node identification (by both surgeons and pathologists), lymph nodes downstaging/regression (cN+/ypN0), and surgeon relativization of lymphadenectomy’s role in these settings have been ascribed as potential reasons for this [46,47]. Second, D1 contamination and D2 noncompliance have been reported in up to 8% and 50% of patients, respectively [48]. These should be regarded as potential confounders and hypothetical causes of overestimation or underestimation of the effect of lymphadenectomy. Third, spleen and distal pancreas resection were specified in the MRC and Dutch trial protocols to achieve complete D2 lymphadenectomy in patients requiring total gastrectomy with increased postoperative complications and mortality and worsening effect on D2-related survival [49]. Fourth, other than the number of retrieved lymph nodes, the anatomical location of the metastatic lymph node might act as a collateral prognostic indicator of tumor lymphatic diffusion and survival. Specifically, it has been shown that the presence of lymph node metastases in stations 4d and 6 is independently associated with an increased risk of concomitant tumor sprouting in station 14v, whereas lymph node metastasis in far-extragastric stations (No. 10–12) has been reported to be associated with worse survival compared to perigastric stations (No. 1–6) and near-extragastric stations (No. 7–9), regardless of cancer location [6,50,51,52]. Fifth, the operating surgeon learning curve, proficiency, and hospital caseload were not detailed, whereas inadequate pre-trial D2 training has been reported with a potential effect on postoperative complications and mortality [53,54,55]. Finally, none of the studies reported the utilization of fluorescent lymphography-guided lymphadenectomy with indocyanine green (ICG). Endoscopic peritumoral injection the day before surgery has been shown to be useful for draining lymph node mapping and nodal identification during minimally invasive resections [56]. ICG mapping has been shown to be associated with enhanced lymphadenectomy standardization and a higher number of dissected nodes alongside a positive effect on survival [57,58]. 

The main strength of our meta-analysis is the appraisal of long-term survival in D2 vs. D1 using the RMSTD. The RMSTD has gained increasing acceptance in oncology as it is a robust and interpretable tool to assess clinical survival benefit of a specific treatment. It matches the area under the survival curves and is easier to interpret compared to HR and RR that may be misinterpreted because both assume a constant risk during follow-up. We recognize that our study is limited by selection/allocation bias and heterogeneity (i.e., demographics, comorbidities, nutritional status, extent of gastric resection, etc.). Oncologic data (i.e., staging, histology, genomics, microsatellite instability, dMMR/MSI, HER-2 expression, PD-L1 testing, and immune infiltration) were not reported [59,60,61]. Finally, our results may not be generalized because of different epidemiologies, genomic characterizations, biomolecular patterns, and correction for early (30-day vs. 90-day) mortality. 

## 5. Conclusions

This study shows that D2 lymphadenectomy seems to be associated with a clinical trend toward improved OS, CSS, and DFS compared to D1. Specifically, the RMSTD analysis shows a mean survival benefit of 1.8, 1.2, and 0.8 months at 60-month follow-up, respectively. Caution is recommended to avoid overestimation of the D2 effect since the clinical benefit of a more extended lymphadenectomy may be a consequence of diverse tumor genomics and molecular profiles. Further studies are mandatory to investigate, whereas a personalized lymphadenectomy extent based on staging, neoadjuvant treatments, tumor biomolecular patterns, and patient’s characteristics might be beneficial. 

## Figures and Tables

**Figure 1 cancers-16-00424-f001:**
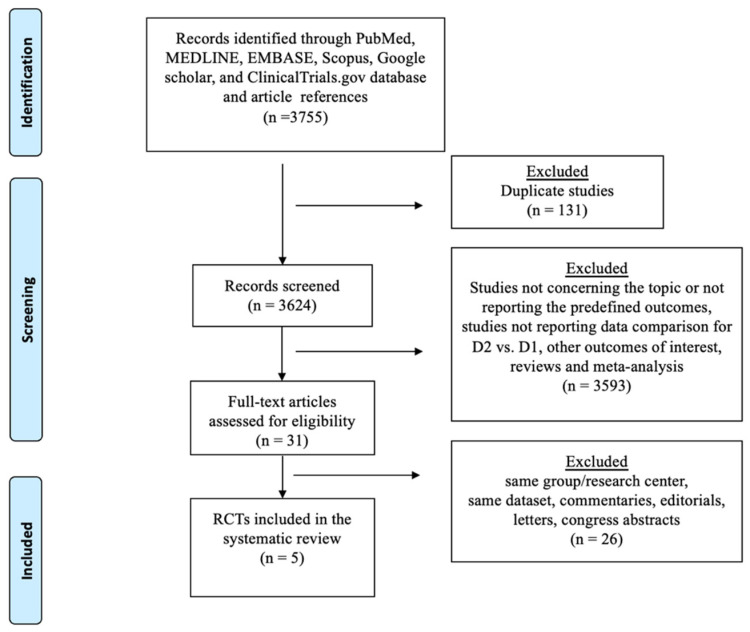
The Preferred Reporting Items for Systematic Reviews checklist (PRISMA) diagram.

**Figure 2 cancers-16-00424-f002:**
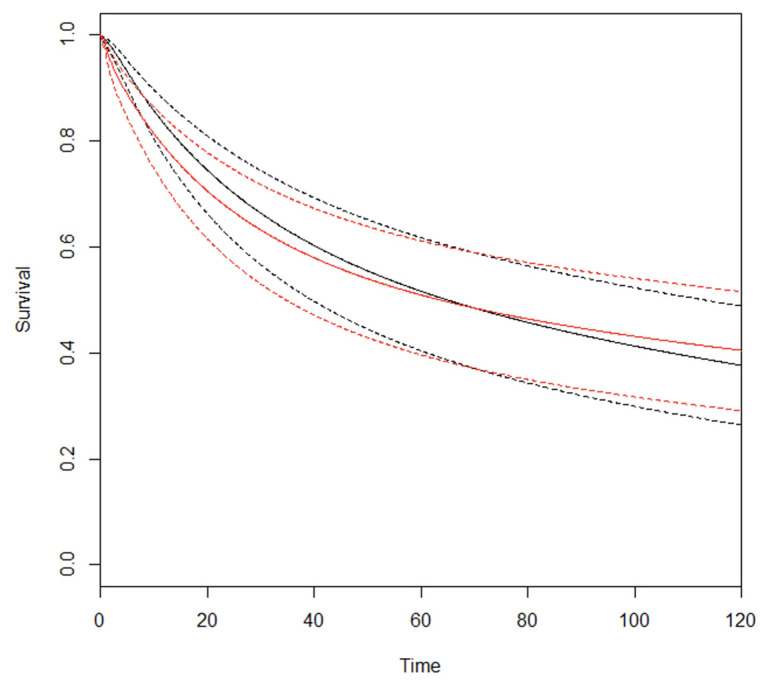
Estimated pooled OS (Y axis) for D2 (red line) and D1 lymphadenectomy (black line). Time (X axis) is expressed in months.

**Figure 3 cancers-16-00424-f003:**
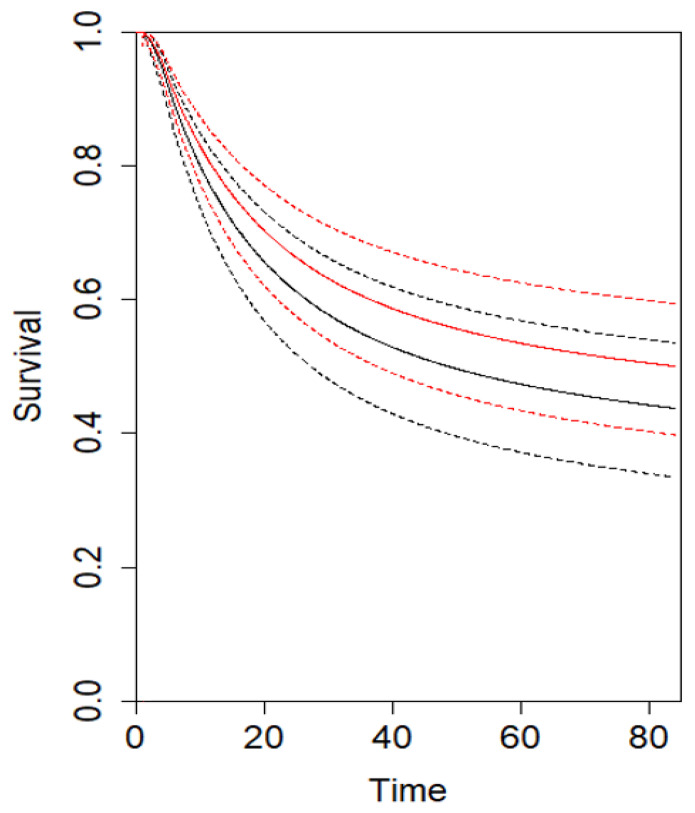
Estimated pooled CSS (Y axis) for D2 (red line) and D1 lymphadenectomy (black line). Time (X axis) is expressed in months.

**Figure 4 cancers-16-00424-f004:**
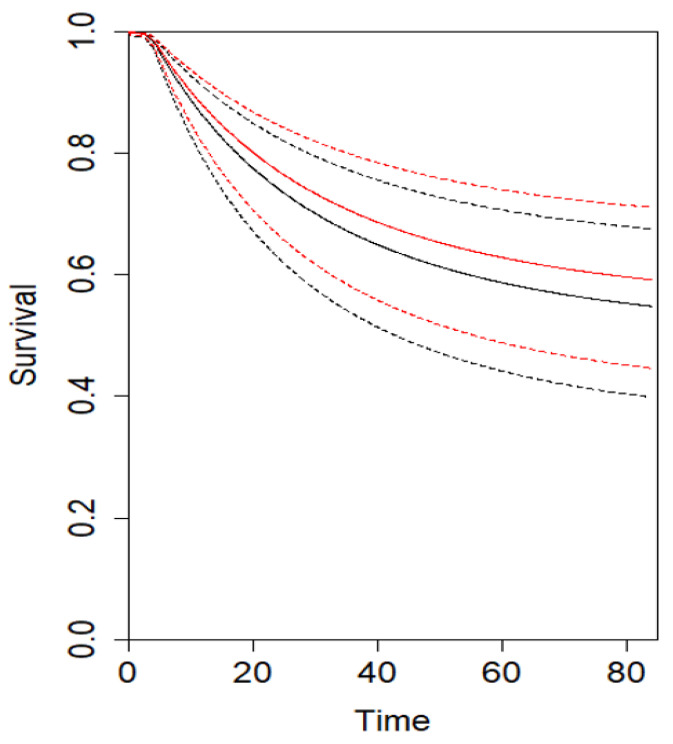
Estimated pooled DFS (Y axis) for D2 (red line) and D1 lymphadenectomy (black line). Time (X axis) is expressed in months.

**Table 1 cancers-16-00424-t001:** Demographic, clinical, and operative data for patients undergoing D2 and D1 lymphadenectomy for gastric cancer. Pathologic tumor stage is reported according to the to the 5th edition of the American Joint Committee on Cancer (AJCC). Yrs: years; M: male; F: female; SG: subtotal gastrectomy; TG: total gastrectomy. Data are reported as numbers, mean ± standard deviation, and median (range).

Author, Year, Trial Tile (Country)	Period	Group	No. Pts	Age (yrs)	M/F	Stage I	Stage II	Stage III	Stage IV	T1-T2	T3-T4	N0	N+	SG	TG
Robertson et al., 1994 Hong Kong trial, (Hong Kong) [30]	1987–1991	D1	25	60 (32–75)	20/5	nr	nr	nr	nr	17	8	14	11	25	0
		D2	29	58 (31–75)	22/7	nr	nr	nr	nr	17	12	14	15	0	29
Cuschieri et al., 1999, MRC trial (UK) [31]	1986–1993	D1	200	67 (38–86)	132/68	67	37	80	0	111	84	69	115	88	110
		D2	200	67 (26–83)	138/62	63	53	75	0	109	86	78	114	91	108
Wu et al., 2006, Taiwan trial (Taiwan), [32]	1993–1999	D1	110	63 (60.9–65.1)	84/26	nr	nr	nr	nr	49	61	39	71	80	30
		D2	111	65.2 (63.2–67.2)	86/25	nr	nr	nr	nr	49	62	44	67	88	23
Songun et al., 2010, Dutch trial (Netherlands) [33]	1989–1993	D1	380	252 (≤70), 128 (>70)	215/165	172	93	84	28	279	94	171	209	265	115
		D2	331	229 (≤70), 102 (>70)	186/145	141	77	74	36	237	82	144	187	205	126
Degiuli et al., 2021, IGCSG trial (Italy) [34]	1998–2006	D1	133	64 (30–81)	65/64	61	24	36	9	91	40	63	68	98	35
		D2	134	62 (22–87)	64/67	56	33	27	15	94	37	57	74	103	31

**Table 2 cancers-16-00424-t002:** Overall survival. The restricted mean survival time difference (RMSTD) restricted to 120 months at different time horizons for the D2 vs. D1 comparison. SE: standard error; 95% CI confidence intervals; mos: months.

Time Horizon	No. Trials	RMSTD (mos)	SE	95% CI	*p* Value
12 months	5	0.2	0.25	−0.72, 0.3	0.37
24 months	5	0.6	0.3	−1.8, 0.5	0.27
36 months	5	1.2	0.9	−3.1, 0.6	0.19
48 months	5	1.7	1.2	−4.1, 0.6	0.15
60 months	5	1.8	1.2	−4.2, 0.7	0.14
72 months	4	1.8	1.5	−4.7, 1.2	0.25
96 months	3	1.8	2.5	−6.6, 3.2	0.48
120 months	3	1.6	3.9	−9.2, 6.1	0.68

**Table 3 cancers-16-00424-t003:** Cancer-specific survival (CSS). The restricted mean survival time difference (RMSTD) restricted to 120 months at different time horizons for the D2 vs. D1 comparison. SE: standard error; 95% CI confidence intervals; mos: months.

Time Horizon	No. Trials	RMSTD (mos)	SE	95% CI	*p* Value
12 months	3	0.1	0.2	−0.3, 0.6	0.48
24 months	3	0.1	0.7	−1.5, 1.4	0.87
36 months	3	0.2	1.3	−2.8, 2.3	0.84
48 months	3	0.1	1.9	−3.6, 3.8	0.95
60 months	3	1.2	2.5	−3.9, 5.7	0.72
72 months	3	1.7	3.1	−4.3, 7.7	0.57
96 months	2	3.7	4.5	−5.1, 12.5	0.41
120 months	2	6.1	5.9	−5.2, 17.7	0.28

**Table 4 cancers-16-00424-t004:** Disease-free survival (DFS). The restricted mean survival time difference (RMSTD) restricted to 120 months at different time horizons for the D2 vs. D1 comparison. SE standard error; 95% CI confidence intervals; mos months.

Time Horizon	No. Trials	RMSTD (mos)	SE	95% CI	*p* Value
12 months	3	0.2	0.12	−0.02, 0.4	0.07
24 months	3	0.4	0.2	−0.2, 0.9	0.17
36 months	3	0.4	0.5	−0.7, 1.4	0.49
48 months	3	0.4	1.0	−1.5, 2.4	0.65
60 months	3	0.8	1.3	−1.7, 3.4	0.53
72 months	3	1.3	1.6	−1.8, 4.5	0.41
96 months	2	2.4	2.3	−2.2, 7.1	0.32
120 months	2	3.5	3.3	−3.1, 10.2	0.29

## Data Availability

Data generated at a central, large-scale facility are available upon request from the corresponding author.

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
