# Peer review of "Long-Term Impact of D2 Lymphadenectomy during Gastrectomy for Cancer: Individual Patient Data Meta-Analysis and Restricted Mean Survival Time Estimation"

_cancers, 2024, doi:10.3390/cancers16020424_

Round 1
Reviewer 1 Report
Comments and Suggestions for Authors
The authors have made a well-written manuscript regarding the Long-Term Impact of D2 Lymphadenectomy During Gastrectomy for Cancer.
I have very few comments since the manuscript overall is well written and presented. The selected references are relevant and include all necessary published data. Tables and figures are likewise of relevant design and content.
In order for lymphadenectomy to be performed successfully, laparoscopic gastrectomy with near-infrared ICG camera is often mentioned. Briefly describe this in the discussion, quoting the reference below.
Park SH, Berlth F, Choi JH, Park JH, Suh YS, Kong SH, Park DJ, Lee HJ, Yang HK. Near-infrared fluorescence-guided surgery using indocyanine green facilitates secure infrapyloric lymph node dissection during laparoscopic distal gastrectomy. Surg Today. 2020 Oct;50(10):1187-1196.
Kwon IG, Son T, Kim HI, Hyung WJ. Fluorescent Lymphography-Guided Lymphadenectomy During Robotic Radical Gastrectomy for Gastric Cancer. JAMA Surg. 2019 Feb 1;154(2):150-158.
Author Response
A dedicated paragraph has been added in the discussion section with appropriate references as requested. Thank you for your positive comments and the possibility to improve our manuscript.
Reviewer 2 Report
Comments and Suggestions for Authors
In the debate about the extent of lymph node dissection in gastric cancer surgery, the superiority of D2 has become unassailable due to the final results of the Dutch trial, the technical perfection of laparoscopic D2 gastrectomy, and the evidence of non-inferiority in recent RCTs of LADG D2. However, there are still few meta-analyses of RCTs, and this is a fascinating paper that casts some doubt on the superiority of D2 and is therefore worth publishing.
However, there are problems with the following points.
1. The red line in Figures 3 and 4 is considered D2, but in Figure 2, the lines are interchanged; the red line is D1, and the black line is D2, right? This should be checked, and the line types should be standardized.
2. Since this is a meta-analysis of RCTs, the authors' statement that "the slight benefit of a more extended lymphadenectomy may be consequence of tumor biology and molecular profile rather than a more extended surgical dissection." is incorrect. As long as the studies were strictly RCTs, the molecular profiles must be identical between D1 and D2. Although the molecular profile is advantageous in molecular targeted therapy, this cannot detract from the superiority of D2 even if the chemotherapy strategy is changed. This is because molecular targeted therapy and surgical therapy cannot be contradictory factors, even if additive effects are obtained. The authors need to modify their discussion and conclusion.
3. Although the superiority of D2 becomes apparent after 60 months in the OS curve in Figure 2, the superiority of D2 is immediately obvious in the estimated pooled CSS curve in Figure 3 and the estimated pooled DFS curve in Figure 4. This difference cannot be explained by stage migration or confounders. This difference is likely due to the technical difficulty and surgical invasiveness of D2. Therefore, with adequate training of LADG D2, the superiority of D2 will become unassailable.
Comments on the Quality of English LanguageEnglish writing is clear and has no problem.
Author Response
We agree with you
1. Figure 2 has been corrected. This was a typo. Sorry for the inconvenient.
2.3 The improved OS, CSS and DFS observed in D2 compared to D1 is limited since at 60-month follow-up D2 patients tended to live 1.8 months, 1.2 months, and 0.8 months more compared to D1. Despite this advantage is unassailable this improved clinical effect should, in our opinion, shed the light on the "real-world" impact of D2. The conclusions section both in the abstract and manuscript have been tempered.
We should notice that molecular tumor profiles and genomics were not assessed in the included studies. Since we cannot certainly assume that there was an equal distribution of genomic patterns among D2 and D1 group our conclusions should be cautious.
In our cautionary and humble opinion further studies are mandatory to give emphasis onto tumor genomics and their impact on multimodal treatments response and survival (doi.org/10.1038/s41568-021-00412-7).
Thank you for your comments and the possibility to improve our manuscript